# A Bosniak III Cyst Unmasking Tubulocystic Renal Cell Carcinoma in an Adolescent: Management with Selective Arterial Clamping and Robotic Enucleation

Marcello Della Corte [1,2,*,†], Elisa Cerchia [2,†], Marco Allasia [3], Alessandro Marquis [3], Alessandra Linari [4], Martina Mandaletti [1,2], Elena Ruggiero [2], Andrea Sterrantino [1], Paola Quarello [5], Massimo Catti [2], Franca Fagioli [5], Paolo Gontero [3] and Simona Gerocarni Nappo [2]

1    Division of Urology, Department of Oncology, School of Medicine, San Luigi Gonzaga Hospital, University of Turin, Regione Gonzole 10, 10043 Orbassano, Italy
2    Division of Pediatric Urology, Regina Margherita Hospital, 10126 Turin, Italy; simogero67@gmail.com (S.G.N.)
3    Department of Urology, Città della Salute e della Scienza, Molinette University Hospital, Corso Bramante 88, 10126 Turin, Italy; paolo.gontero@unito.it (P.G.)
4    Department of Pathology, Azienda Ospedaliera Città della Salute e della Scienza, 10126 Turin, Italy; linaralessandra@libero.it
5    Division of Onco-Hematology, Department of Pediatrics and Pediatric Specialties, Regina Margherita Children Hospital, 10126 Turin, Italy
\*    Correspondence: dellacortemarcello@gmail.com; Tel.: +039-011-902-6477
†    These authors contributed equally to this work.

**Abstract:** The Bosniak classification of renal cysts aims to provide a probabilistic risk assessment indicating the likelihood of malignancy from imaging findings. Originally designed to classify adult renal cysts based on computed tomography findings, the Bosniak classification has been extended to pediatric patients, with some adjustments made with the aim of accommodating magnetic resonance imaging (MRI) and ultrasonography (US). Bosniak IV lesions are rare in adolescents, indicating localized renal cell carcinoma and requiring surgical intervention. In contrast, Bosniak III lesions can be treated conservatively, although there is a lack of specific guidelines on their management. We present a case of a 14-year-old boy with a Bosniak III lesion, which was incidentally detected during the US evaluation of a left varicocele. After a 12-month follow-up, MRI revealed progression to a Bosniak IV cyst. Robot-assisted tumor enucleation was performed with selective artery clamping when the patient was 15. Histopathology showed tubulocystic renal cell carcinoma without adverse features. Immunocytochemistry supported a favorable prognosis of this rare tumor (<1% of renal tumor), thus obviating the need for adjuvant treatment. At the 18-month follow-up, no recurrence or distant metastasis were observed. This case highlights the importance of an aggressive treatment in persistent Bosniak III and Bosniak IV renal cysts in children and adolescents and the necessity to offer a nephron-sparing surgery.

**Keywords:** Bosniak III cyst; renal cyst; pediatric; tubulocystic renal cell carcinoma; cystic neoplasm of kidney; renal cell carcinoma; nephron-sparing surgery; robotic surgery; partial nephrectomy

## 1. Introduction

Renal cysts represent an infrequent finding in the pediatric population, with an estimated overall incidence ranging from 0.2% to 5%. While their prevalence in adults correlates with advancing age, pediatric renal cysts lack a discernible causative factor. In children, a renal cyst may remain isolated and stable over time or may precede the onset of cystic kidney disease, chronic kidney disease, symptomatic enlargement, and malignant transformation [1]. Therefore, defining the nature and the natural history of renal cysts is of utmost importance to choose the most appropriate management strategy.

The Bosniak classification was developed in the adult population to ascertain the risk of malignancy of renal cysts based on computed tomography (CT) characteristics [2]. Bosniak I and II cysts are deemed benign lesions that do not necessitate a follow-up [3] but may require treatment when symptomatic, presenting with discomfort, flank pain, hematuria, or a palpable mass. Intervention in these cases varies from minimally invasive procedures, like percutaneous puncture with or without sclerotherapy, to open, laparoscopic, or robot-assisted surgery with options including cyst marsupialization, cyst resection, nephron-sparing surgery (NSS), or total nephrectomy [4].

In contrast, Bosniak IV cysts are predominantly malignant tumors exhibiting pseudo-cystic changes, and their surgical excision is suggested, preferring NSS when possible rather than radical nephrectomy [3].

Bosniak IIF and III cysts present significant challenges to clinicians. Distinguishing benign from malignant tumors in categories IIF/III relies on imaging, primarily CT, with magnetic resonance imaging (MRI) and contrast-enhanced ultrasounds (USs) taking on expanding roles. The EAU guidelines recommend either managing Bosniak type III cysts similarly to localized renal cell cancer (RCC) or considering active surveillance (AS) with a 'weak' strength rating. On one hand, surgical intervention for Bosniak III cysts can often result in the over-treatment of tumors with low malignant potential. On the other hand, considering the generally favorable outcomes for these patients, a surveillance approach is a valid alternative [3].

Unlike the adult population, there is presently no consensus on the most effective approach for renal cysts in children. Recent studies advocate the classification of pediatric renal cysts using a US-based modified Bosniak (mBosniak) classification system [5]. This approach aims to decrease ionizing radiation exposure by circumventing unnecessary CT studies, minimizing extended surveillance and related morbidity, and reducing the frequency of surgical interventions for benign lesions [1]. Cystic renal lesions designated as Bosniak classes I-II demonstrate a benign character. On the other hand, lesions classified as classes III-IV are more inclined to indicate intermediate or malignant pathology, thus necessitating consideration for surgical excision [6].

Presently, despite existing evidence pointing towards malignancy in pediatric Bosniak III cysts, there is still a lack of consensus on their management [1].

In this study, we describe our experience in an adolescent with a Bosniak III cyst, which was managed with robot-assisted minimally invasive NSS, and revealing a tubulocystic renal cell carcinoma (TCRC).

The Case Reports (CARE) guidelines have been followed to enhance the accuracy, transparency, and utility of this study [7].

## 2. Case Report

A 14-year-old boy with a left varicocele underwent an abdominal US (not recommended by the EAU/ ESPU guidelines), which incidentally revealed a hyperechoic round mass in the lower pole of the left kidney measuring $35 \times 33$ mm. Abdominal low-dose CT confirmed a partially exophytic lesion in the lower pole of the left kidney. The lesion exhibited a hypodense lower part suggestive of liquid content without contrast enhancement and a solid upper part with point-like calcifications and contrast enhancement. It was classified as a Bosniak III cyst (Figure 1a,b).

The patient, who was initially managed conservatively at a peripheral center, was referred to our tertiary center after 12 months of follow-up due to the persistence of the lesion and case management.

The patient had a good general health condition, reporting an occasional sensation of heaviness at the left testicle, and was otherwise asymptomatic. Neither he nor his parents reported previous hematuria, flank pain, weight loss, or fatigue. His blood pressure was within the normal range. During the clinical examination, the abdomen was flat, soft, and non-tender with regular bowel sounds on auscultation, and there were no palpable masses,

confirming the clinical finding of grade II left varicocele. His past medical history was negative for significant pathological events.

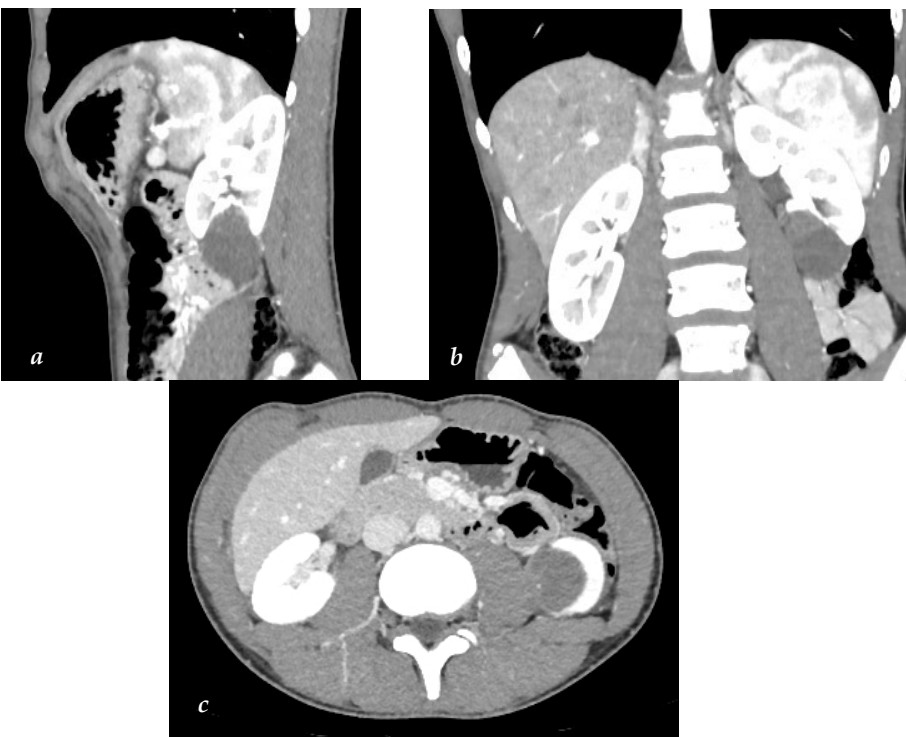

**Figure 1.** (**a**–**c**) Abdominal CT images. (**a**) Sagittal view of the cleavage plane between the cyst and renal parenchyma conceivable. (**b**) Coronal view of tumor relationships with adjacent organs. (**c**) Transverse view with no signs of invasion of the surrounding organs.

MRI was planned and, during the creatinine level measurement for the administration of an MRI contrast, complete blood count and electrolyte tests were requested, and they were within normal ranges. The MRI revealed a partially solid upper portion and a fluid-filled lower portion of the lesion in both T1 and T2 phases, resulting in the upgrade of the Bosniak classification from III to IV (Figure 2). A chest CT scan did not exhibit any suspected findings.

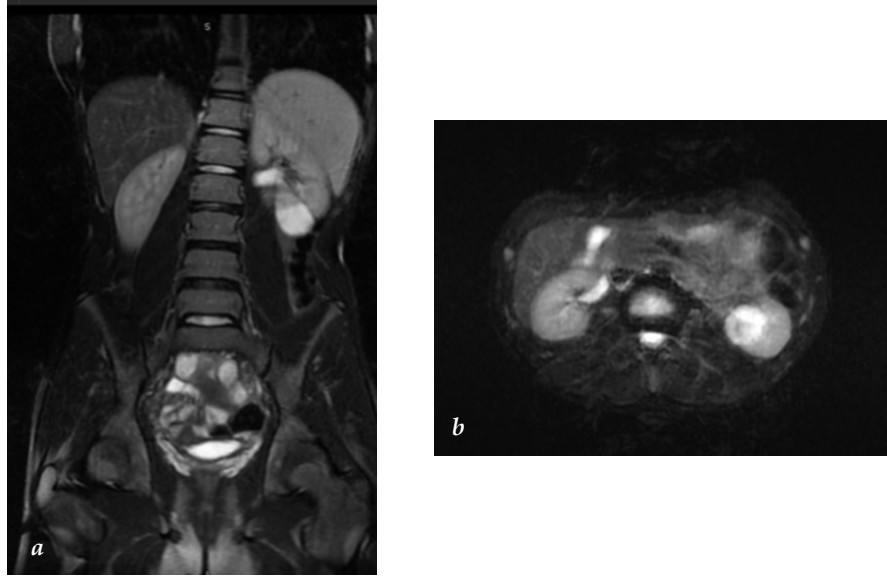

**Figure 2.** (**a**,**b**) Abdominal MRI: the coronal and transverse views exhibiting the tumor position and excluded lymph node localization.

After a case discussion at multidisciplinary oncologic meetings, upfront surgery was planned. A three-dimensional (3D) reconstruction of the mass was performed (Medics Srl©, Moncalieri, Turin, Italy) (Figure 3a–c). At the age of 15, the boy underwent robot-assisted tumor enucleation. According to the UMBRELLA-SIOP Protocol, hilar and para-aortic lymphadenectomy were also performed [8,9]. During NSS, thanks to the 3D virtual reconstruction, only the inferior branch of the renal artery supplying the tumor was clamped using a single laparoscopic bulldog clamp (Aesculap, Tuttlingen, Germany). Indocyanine green injection (ICG) confirmed proper tumor ischemia, with well-preserved perfusion of the upper two-thirds of the kidney (Figure 4a–c).

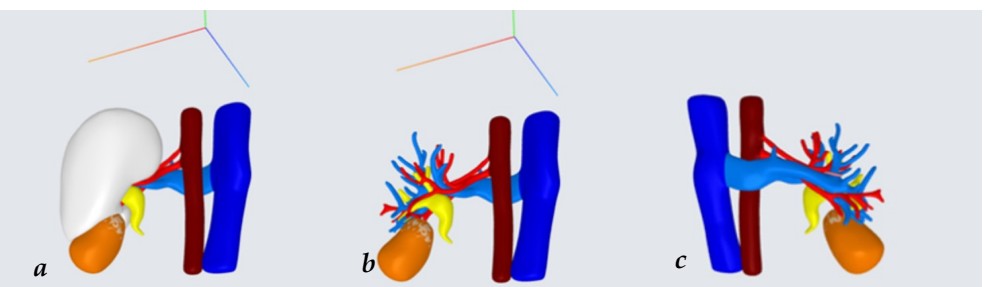

**Figure 3.** (**a**–**c**) Virtual 3D reconstruction. (**a**) Posterior view; (**b**) posterior view where kidney is excluded; (**c**) anterior view of tumor. Legend: tumor = orange, kidney = white, arterial vessels and aorta = red, venal vessels and vein cava = blue.

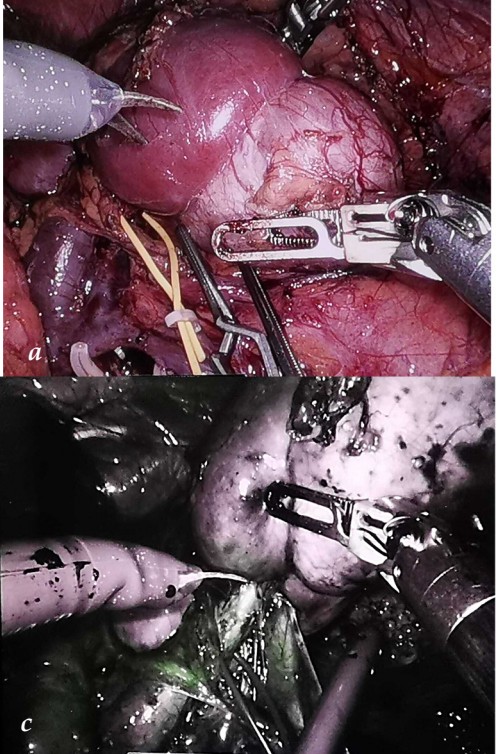

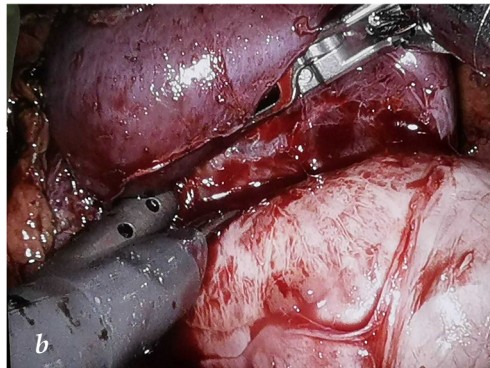

**Figure 4.** (**a**–**c**) Intraoperative view. (**a**) First appearance of tumor; (**b**) enucleation beginning following pseudocapsule plane; (**c**) ICG test confirming proper tumor and lower pole ischemia.

The operative time was 205 min, including the docking time (20 min), with selective artery clamping for 22 min. No intraoperative complications occurred, and the enucleation was complete (Figure 5a,b). The postoperative period was uneventful, and the patient was discharged on the third postoperative day.

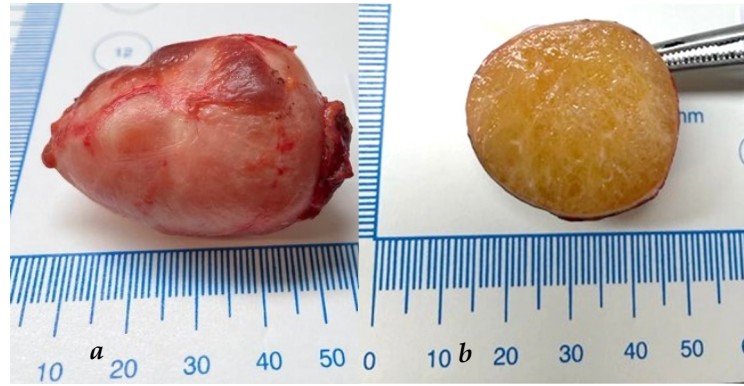

**Figure 5.** (**a**,**b**) The enucleated tumor. (**a**) The tumor exhibits a well-respected pseudo capsule; (**b**) the yellowish homogeneous content after the sagittal section.

Histopathology revealed a TCRC (ISUP-WHO 2016) without capsule involvement, lymph node metastasis, or vascular invasion, staged as pT1aN0R0. Immunocytochemistry displayed positive staining for PAX8, CK18, CK7, and CD10; estrogenic and progesterone receptors were not expressed. No mitotic activity was observed. Based on histology (Figure 6a–d), no adjuvant treatment was deemed necessary, and a strict follow-up was planned. At the 18-month follow-up, no signs of local recurrence or distant metastasis were evident.

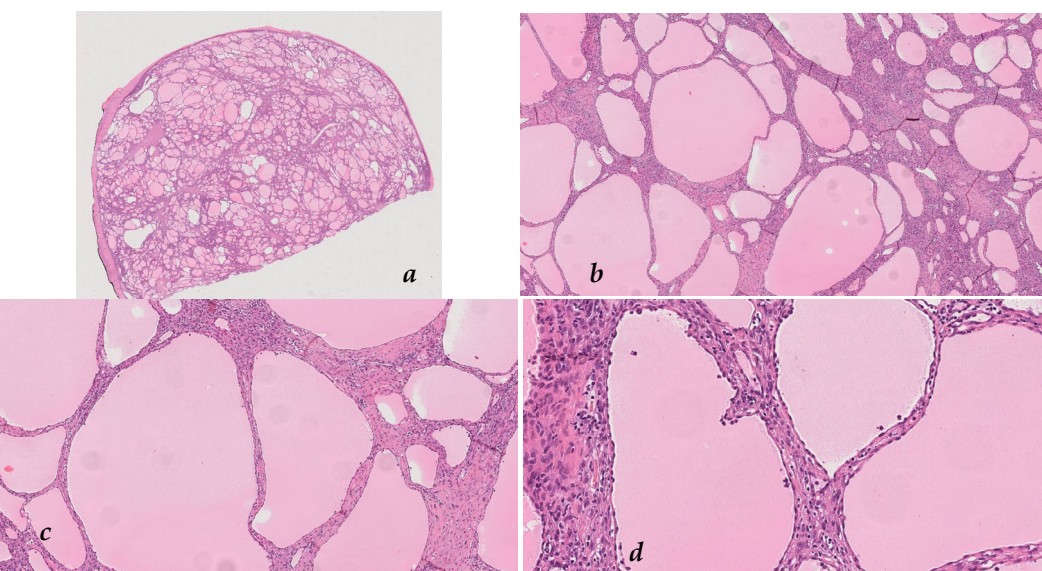

**Figure 6.** (**a**–**d**) Histopathology images. Cuboidal and columnar cells (**c**,**d**) in a fibrotic stroma with cysts and small tubules (**a**–**d**). The cells have few mitotic figures but prominent nucleoli (**d**). We offer different levels of magnification to better appreciate these characteristics: (**a**) 10×; (**b**) 40×; (**c**) 80×; (**d**) 100×.

## 3. Discussion

The Bosniak classification system of renal cysts is based on the CT and MRI findings. A new US-based classification has only recently been proposed, where the size of renal cysts can be used to distinguish benign from malignant cysts with elevated sensitivity and specificity [5]. Despite strict imaging classification, neither official guidelines nor recommendations exist for pediatric patients. In particular, the management of Bosniak III cysts is still debated, as a consensus is still lacking, and represents a significant challenge for clinicians.

Although the EAU guidelines allow for conservative management for adults, this type of approach is not preferable in pediatrics, where the risk of malignancy still subsists, ranging from 44% to 58% [3]. It must be kept in mind, however, that upfront surgery could increase perioperative morbidity, especially with unnecessary surgery performed for benign lesions.

In a recent study, Wang et al. surveyed pediatric urologists to determine their preferred treatment modality for an asymptomatic patient presenting with a renal cyst. The predominant choice for managing a symptomatic child with a simple renal cyst was laparoscopic marsupialization (39%). Conversely, the responses exhibited greater diversity when asked to determine the preferred treatment for a CT-identified Bosniak III cyst, with laparoscopic cyst resection (26%) and partial or total nephrectomy (19%) emerging as the most frequently selected options [1].

In the present case, the increase in the Bosniak score dictated surgical planning. However, MRI played a crucial role in offering a 'complementary' depiction of the cyst. In instances of uncertainty or when an expectant approach (active surveillance) is not justified, NSS may represent a preferable option, finding a balance in the curative approach without compromising the kidney function. Partial nephrectomy has been shown to effectively preserve kidney function after surgery, potentially lowering the risk of cardiovascular disorders in the long term. Comparative analyses with radical surgical approaches, drawn from multiple retrospective studies on extensive databases, have suggested a potential reduction in cardiovascular-specific mortality and improved overall survival for partial nephrectomy compared to radical nephrectomy for small renal tumors (T1). Additionally, some series indicate that these benefits may be more prominent in younger patients, such as the presented case, and/or those without significant comorbidities at the time of surgical intervention [3].

Recently, the integration of 3D printing has played a crucial role in preparing surgeons for NSS, facilitating an optimal pre-operative understanding of the surgical dissection plans [10]. Preoperative imaging is crucial for the development of 3D virtual models and subsequent surgical planning. Utilizing CT and MRI imaging, 3D reconstruction improves the topographic visualization of the tumor and adjacent structures, effectively reducing the risk of inadvertent damage to vessels and surrounding organs [11].

TCRC is an exceptionally uncommon variant of RCC, which is typically observed in adults, with an anecdotal case reported in an older adolescent [12]. The World Health Organization (WHO) officially included it in the renal cell carcinoma classification in 2016. This tumor tends to affect male patients, with a male-to-female ratio of 7:1, but specific risk factors for this tumor have not been identified yet. While some patients may experience abdominal distention, pain, or hematuria [13], most patients are asymptomatic at presentation, and the lesion is an incidental finding at imaging.

Most of these tumors typically display an indolent behavior with low metastatic potential; however, isolated cases of local recurrence and metastases to various sites have been reported. The preoperative diagnosis of TCRC poses a challenge. Macroscopically, the tumor exhibits well-defined, non-encapsulated features and commonly manifests a cystic component (Bosniak type III or Bosniak type IV). The dissected surface appears spongy, with a coloration ranging from white to gray, often resembling 'bubble wrap' [14].

If a solid portion is present within the renal mass, TCRC is considered. However, when only a cystic component is evident, distinguishing between TCRC and renal cysts becomes challenging. Identifying contrast enhancement on CT scans is crucial but remains challenging due to the low vascularization of TCRC and the limited presence of solid tissue components. MRI is valuable to reveal the microcystic nature of these tumors, benefiting from its superior contrast resolution. Additionally, US has proven to be useful in identifying TCRC, displaying high echogenicity and posterior acoustic enhancement due to its multicystic characteristics separated by multiple thin septa [14,15].

Surgical intervention is recommended due to the tumor's indolent characteristics and low propensity for metastasis [3,16]. Timely diagnosis and suitable intervention have the

potential to impede cancer dissemination, decelerate disease advancement, and enhance patient recovery [17].

The recommendation for TCRC is for radical nephrectomy, with NSS remaining a viable option for smaller T1 tumors (<7 cm) [3,13].

In cases where TCRC is associated with a Bosniak I or Bosniak II cyst, marsupialization of the cyst leads to upstaging and local recurrence and renders the condition no longer manageable with sole surgical treatment, necessitating a systemic approach [18]. Despite a weak strength rating, the EAU guidelines recommend managing Bosniak type III cysts similarly to localized RCC or proposing AS [3]. This results in a wide range of possible therapeutic approaches for the same disease depending on its imaging appearance and has significant implications from both prognostic and medicolegal standpoints in cases of undertreatment. This peculiarity has earned him the nickname 'the great imitator' [14].

To our knowledge, this case represents the first instance of TCRC diagnosed in a pediatric patient and managed robotically with selective arterial clamping. Despite the limited 18-month follow-up, this approach proved to be successful, resulting in a secure and effective tumor enucleation. The success of the presented case is intricately linked to robotic NSS, which was made possible by the collaboration of adult and pediatric urologic surgical teams. Considering the potential oncological nature of the lesions, collaboration with adult urologists experienced in daily robotic NSS practice for RCC was deemed crucial. The favorable tumor location at the lower renal pole significantly facilitated the NSS approach, preventing the loss of healthy renal parenchyma. Lastly, the utilization of 3D reconstruction and intraoperative ICG allowed for the selective clamping of a secondary arterial branch, minimizing the impact of warm ischemia on renal function [10].

Given the extended life expectancy of patients with T1 tumors and the heightened risk of long-term complications, such as hypertension and chronic renal failure following radical nephrectomy, NSS should be the preferred surgical option whenever feasible and oncologically safe. The integration of 3D preoperative imaging reconstruction and robot-assisted surgery, particularly in the hands of experienced practitioners, possibly in collaboration with adult urologists, emerges as a valuable tool that will further enhance the feasibility of NSS in pediatric oncology patients in the future [11].

## 4. Conclusions

Renal cysts pose a challenging scenario in pediatric urology given their rarity and the lack of official guidelines for their management. Surgical intervention is advisable for persistent Bosniak III and Bosniak IV lesions due to their elevated risk of malignancy. TCRC is a rare tumor, and its clinical presentation is typically silent.

The robot-assisted approach is a safe and precise surgical option for suspected renal cysts, providing both minimal invasiveness and NSS. According to the literature, this represents the first reported case of a young adolescent treated for TCRC with robot-assisted enucleation.

**Author Contributions:** Conceptualization, M.D.C., E.C., M.A., A.M., M.M., E.R., A.S., P.Q., M.C., F.F., P.G. and S.G.N.; methodology, M.D.C., E.C., A.M., A.L., P.Q., F.F., P.G. and S.G.N.; software, M.D.C., E.C., A.M., P.Q., F.F., P.G. and S.G.N.; validation, M.D.C., E.C., A.M., P.Q., F.F., P.G. and S.G.N.; formal analysis, S.G.N.; investigation, M.D.C., E.C., A.L. and S.G.N.; resources, M.D.C., P.G. and S.G.N.; data curation, M.D.C. and S.G.N.; writing—original draft preparation, M.D.C., E.C., M.A., A.M., A.L., M.M., E.R., A.S., P.Q., M.C., F.F., P.G. and S.G.N.; writing—review and editing, M.D.C., E.C., M.A., A.M., A.L., M.M., E.R., A.S., P.Q., M.C., F.F., P.G. and S.G.N.; visualization, M.D.C., P.G. and S.G.N.; supervision, S.G.N.; project administration, M.D.C. and S.G.N.; funding acquisition, M.D.C. All authors have read and agreed to the published version of the manuscript.

**Funding:** This research received no external funding.

**Institutional Review Board Statement:** Not applicable.

**Informed Consent Statement:** Written informed consent was obtained from the patient's parents to publish this paper.

**Data Availability Statement:** No new data were created. The clinical data are available upon request in anonymous form due to privacy restrictions.

**Conflicts of Interest:** The authors declare no conflicts of interest.

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
