# Peer review of "A Bosniak III Cyst Unmasking Tubulocystic Renal Cell Carcinoma in an Adolescent: Management with Selective Arterial Clamping and Robotic Enucleation"

_2673-4095, doi:10.3390/surgeries5020034_

Round 1
Reviewer 1 Report
Comments and Suggestions for Authors
The study reported a case about tubulocystic renal cell carcinoma diagnosed in a pediatric patient and treated by selective arterial clamping and robotic enucleation. In general, it is meaningful to some extent, but still has some parts to be improved. My comments are as follows,
1. The symptoms, the results of other preopreative examinations and the past medical history of this patient need to be provided.
2. The transverse position of abdominal CT and MRI need to be provided.
3. The histopathology images should be added to make sure the accuracy and authenticity of this case.
4. The scale, font size and resolution of each figure should be adjusted.
5. A thorough revision of the manuscript by an expert is necessary to ensure that the special case is clearly communicated to the reader through correct grammar, spelling, and sentence structure.
Comments on the Quality of English LanguageMinor editing of English language is required.
Author Response
Dear Reviewer,
We sincerely appreciate your time and effort in revising our manuscript, as well as your valuable comments. We aim to address each point raised, with the hope that the current version meets the standards necessary for manuscript publication.
Q: 1. The symptoms, the results of other preopreative examinations and the past medical history of this patient need to be provided.
A: 1. Thank you for your suggestion. We have added a proper paragraph between lines 104 and 113.
Q: 2. The transverse position of abdominal CT and MRI need to be provided.
A: 2. We have included the requested transverse views for both CT and MRI.
Q: 3. The histopathology images should be added to make sure the accuracy and authenticity of this case.
A: 3. We acknowledge the importance highlighted by the Reviewer regarding the inclusion of histopathology images for authenticity. We have added the suggested images accordingly. Thank you for bringing this to our attention.
Q: 4. The scale, font size and resolution of each figure should be adjusted.
A: 4. We appreciate your feedback. The images have been rearranged, and the resolution is now at 600 dpi, surpassing the requirement outlined by MDPI guidelines. Unfortunately, due to limitations with our hospital's older radiology systems, further improvements in image quality are not feasible.
Q: 5. A thorough revision of the manuscript by an expert is necessary to ensure that the special case is clearly communicated to the reader through correct grammar, spelling, and sentence structure.
5. A language revision has been completed.
As this was the last reply, we would like to express our gratitude once again for your invaluable input in enhancing our work.
Best Regards,
Reviewer 2 Report
Comments and Suggestions for Authors
The authors present a very interesting clinical case of a 14-year-old adolescent patient with a Bosniak III renal cyst that progressed to IV and required surgical treatment.
The introduction is adequate, concise and well oriented for the presentation of the clinical case.
The writing of the case is also correct, but I miss some intraoperative images of the cyst. The images of the specimen are of good quality, but the intraoperative image, also robot-assisted, can be very useful.
The discussion of the article is adequate, highlighting the importance of close follow-up of these patients and early surgical treatment in patients with suspected progression or diagnostic doubt. The clinical applicability of this case is very important for this reason.
The literature provided is fairly recent, with most articles less than 5 years old.
Comments on the Quality of English LanguageThe article is well written and structured, with minor grammatical errors, such as verb tenses or typographical errors. I recommend a revision of the manuscript by a native speaker, for minor editing of English language.
Author Response
Dear Reviewer,
We sincerely appreciate your time, effort, and valuable comments in reviewing our manuscript.
We have now included intraoperative images as per your recommendation. Specifically, we have selected images depicting: (5 a) the initial appearance of the tumor, (5 b) the commencement of tumor enucleation along the plane of its pseudocapsule, and (5 c) the indocyanine test to assess tumor ischemia. We trust that you will find these additions beneficial. Naturally, the image sizes will be adjusted as per the requirements of the MDPI Editorial office.
We once again express our gratitude for your invaluable input in improving our work.
Best Regards,
Round 2
Reviewer 1 Report
Comments and Suggestions for Authors
The authors have addressed all my comments.
Comments on the Quality of English LanguageMinor editing of English language is required.